

# Relations of physical and biogenic reworking of sandy sediments in the southeastern North Sea

Knut Krämer[1], Soeren Ahmerkamp[2,1], Ulrike Schückel[3], Moritz Holtappels[4,2,1], and Christian Winter[5]

[1]MARUM – Zentrum für Marine Umweltwissenschaften, Universität Bremen, Leobener Str. 8, D-28359 Bremen, Germany
[2]Max-Planck-Institut für Marine Mikrobiologie, Celsiusstrasse 1, D-28359 Bremen, Germany
[3]Landesbetrieb für Küstenschutz, Nationalpark und Meeresschutz Schleswig-Holstein (LKN.SH), Schlossgarten 1, D-25832 Tönning, Germany
[4]Alfred Wegener Institute, Helmholtz Centre for Polar and Marine Research, Am Alten Hafen 26, D-27568 Bremerhaven, Germany
[5]Institut für Geowissenschaften, Christian-Albrechts-Universität zu Kiel, Otto-Hahn-Platz 1, D-24118 Kiel, Germany

**Correspondence:** Knut Krämer (kkraemer@marum.de)

**Abstract.** The reworking of sandy sediments in shallow coastal and shelf seas is mainly driven by physical forcing in the form of wave- and current-induced shear stress. As an important habitat for benthic species seeking shelter and food, the upper seafloor is also marked by intense bioturbation. Although this reworking activity is recognized as an important mechanism for the exchange of particulate matter and solutes between sediment and water column, quantifications and assessments of the relative importance of physical and biogenic reworking of subtidal shelf sediments are rare.

This work presents in situ measurements of volumetric reworking rates from six different locations in the southeastern North Sea. The investigated sites cover a range of water depths between 23 and 41 m, different magnitudes of physical (wave and current) forcing and sedimentological conditions as well as different habitats and benthic communities. The measured biogenic reworking rates reach up to 14% of physically driven reworking via bedform migration.

Comparisons with physical quantities water depth, median grain size, bottom water temperature and flow velocity reveal good correlations and allow for an approximation of the biogenic reworking rate from a combination of these readily available oceanographic parameters.

The diffusive relocation of sediment by benthic fauna also influences the topography of small scale bedforms and may reduce their height by up to 10% in a few hours during hydrodynamically inactive conditions.

The observations show that even in an energetic environment such as the southeastern North Sea, the benthic fauna contributes an important regulating ecosystem service by overturning upper seafloor sediments. This reworking mechanism becomes particularly important in areas and during periods of sub-threshold conditions for physically driven sediment reworking.

## 1 Introduction

The seafloor in shallow coastal and shelf seas is an important ecosystem which provides important environmental services such as carbon cycling and nutrient turnover (Marinelli et al., 1998; Chen and Borges, 2009; Snelgrove et al., 2014; Marchant et al., 2016). This energetic environment is characterized by frequent morphodynamic changes, driven by physical forcing due





to currents and waves. Aditionally, the upper sediment layers provide a food source and shelter for a large number of benthic species. They are therefore subject to intense bioturbation resulting from their burrowing and foraging activity (Rhoads and Boyer, 1982; Grant, 1983). This reworking activity of the benthic fauna is well recognized as an important contributor to sediment turnover (Mermillod-Blondin and Rosenberg, 2006; Meysman et al., 2006). Studies investigating the role of bioturbation

cover laboratory experiments on dedicated species (Wrede et al., 2017) or theoretical considerations about effects of biogenic reworking e.g. on bedform dimensions (Soulsby et al., 2012). In situ observations of biologically driven morphodynamics are scarce and often limited to relatively easy accessible intertidal areas (Grant, 1983; Koo and Seo, 2017).

Laboratory measurements of biogenic reworking rates often use tracer techniques (Maire et al., 2008), which in combination with mathematical models allow for the computation of biodiffusion coefficients. However, they do not readily provide

volumetric rates that can be compared with physically driven sediment reworking. A pioneering study on an intertidal sandflat by Grant (1983) with indirect estimation of sand ripple migration showed that biogenic reworking rates may reach up to one third of physically driven bedload transport. In hydrodynamically less active environments such as deep seas or lakes, the role of bioturbation becomes more important (Broach et al., 2016). Bioturbators in the benthic community are therefore regarded as important ecosystem engineers with a regulating function for geobiochemical cycling at the benthic interface (Solan and

Kennedy, 2002).

This study presents in situ measurements of the morphodynamic changes of the micro-bathymetry to physical forcing by currents and waves and bioturbation in exemplary areas in the German Bight. The physical drivers and the morphodynamic effects are linked by the concept of the critical shear stress required for sediment motion. When the combined shear stress from wave and current action exceeds this critical threshold, derived from sedimentological properties, small-scale bedforms

(ripples) form and begin to migrate in the dominant current direction. During periods with hydrodynamic conditions well below the threshold of sediment motion, sediment relocation by benthic fauna is observed, often in characteristic pattern of local erosion and deposition. Volumetric reworking rates are computed as the difference between successive digital elevation models (DEMs) of the sediment surface. The resulting biogenic reworking rates are used to quantify the effect of biogenic reworking in comparison to physical sediment transport.

## 25  2   Study area

### 2.1   Bathymetry and sediment

The data presented here were collected at six locations with characteristic sediment properties in the German Bight in the southeastern North Sea during five cruises with R/V *Heincke* in the scope of projects *NOAH*[1] and *MARUM CCP5* (Fig. 1). Water depths range from 23 m at station CCP-G near Helgoland island to 48 m at station CCP-J in the deep region southeast

of the Dogger Tail End.

---

[1]https://noah-project.de/



The seafloor sediment consists mainly of fine sands with increasing mud content for areas located in the Elbe palaeovalley (Figge, 1981). Small-scale bedforms (ripples) of 1–2 cm height and around 20 cm length are omnipresent in sandy regions in water depth between 20 and 40 m (Krämer and Winter, 2016; Ahmerkamp et al., 2017). Morphological features at the deeper stations with weak bottom currents and beyond the reach of surface waves are of mostly of biogenic origin.

## 2.2 Ranges of physical forcing

The physical forcing in the southeastern German Bight is determined by tidal and wind-driven currents and wave action (Kösters and Winter, 2014). Its local magnitude depends on the water depth, tidal range, phase in the spring-neap-cycle and seasonal meteorological forcing. Bedload sediment transport is driven by shear stresses generated by wave and current action on the seabed ($\tau_w$, $\tau_c$). Their combined effects on the reworking of sediment can be linked with sedimentological properties via the critical shear stress ($\tau_{crit}$), determined from sedimentological characteristics such as median grain size $d_{50}$ and immersed weight $\rho'_s$.

Large-scale pattern of combined wave and current shear stress mainly scale with water depths (Fig. 1). Shallower areas along the North and East Frisian coasts are subjected to stronger wave action than deeper areas. Seasonal differences in meteorological forcing generate waves, high and long enough to affect the seafloor, more frequent during storm events in the winter season (Van der Molen, 2002). Wave-induced shear stresses therefore usually do not reach supercritical conditions for sediment motion from late spring to early fall. The tidal current magnitude and resulting shear stresses are minimal close to the amphidromic point south of Jutland Bank and increase towards the coasts. In shallow areas with water depth smaller than 25 m, current induced shear stresses above the threshold of motion are regularly exceeded for a few hours around peak flood or ebb flow (Krämer and Winter, 2016; Ahmerkamp et al., 2017). Tidal shear stresses additionally vary over the spring-neap cycle, adding a semimonthly timescale determining sub- or supercritical conditions for bedload transport.

Vice versa, this means that for large areas in the southeastern North Sea for a good part of the tidal cycle (current forcing) and a good part of the year (wave forcing), sub-threshold conditions for sediment transport prevail.

## 2.3 Seafloor morphodynamics

Physical reworking by ripple migration is an omnipresent process in the sandy, shallow areas of the German Bight. Active ripples adapt to the direction of tidal flow and migrate in an order of their length scale over a period of a few hours during flood and ebb flow. The activity and migration rate of ripples is mainly controlled by the magnitude of tidal flow, varying over the spring-neap cycle. Although they may not often become the dominant driver in ripple generation, the stirring effect of waves facilitates ripple migration. Areas in the Elbe palaeovalley located in water depths of around 40 m and with higher mud contents do not exhibit active ripples and wave driven sediment transport only occurs during intense storm events.



## 2.4 Benthic communities

The spatial distribution of macrofauna communities in the southeastern North Sea reveales five macrofauna communities, namely the *Tellina (Fabulina) fabula* community, the *Amphiura filiformis* community, the *Nucula nitidosa* community, the *Goniadella spisula* community and the *Bathyporeia spp.* community (Meyer et al., 2018). Spatial distribution of macrofauna communities are in particular structured by environmental parameters such as water depth, sediment structure, tidal forcing and water temperature (Reiss et al., 2009; Meyer et al., 2018).

## 3 Data and Methods

### 3.1 Lander deployments

The data discussed in this study were obtained by an autonomous seafloor observatory (lander) *Lance-A-Lot* (see Ahmerkamp et al. (2017) and supporting information). Most deployments were started during low water slack so as to cover the following flood period. The platform is equipped with a downward-looking acoustic Doppler current profiler (ADCP) to record the vertical profile of near-bed flow velocity and a laser scanning system to record the micro-bathymetry. The available battery capacity allows for an observation period of around 12 hours.

### 3.2 Hydrodynamic forcing

Current-induced shear stresses $\tau_c$ were computed using a logarithmic fit to 10-minute averaged velocity profiles over a range of 1 m from the downward-looking ADCP (*Teledyne RDI Workhorse Rio Grande 1200 kHz*, with a sampling frequency of 1 Hz) following the procedure described by Soulsby (1997, p.53pp). Only data with regression coefficients $R > 0.9$ were used.

Wave induced shear stresses $\tau_w$ were computed using the significant wave height $H_s$ and peak period $T_p$ from a North Sea model (Helmholtz-Zentrum Geesthacht Zentrum für Material- und Küstenforschung GmbH (HZG), 2016). The model results were validated for exemplary periods with waverider buoy measurements recorded near Helgoland island. The maximum combined wave and current shear stress is computed following Soulsby (1997, p.87pp) as

$$\tau_{max} = \sqrt{(\tau_m + \tau_w \cdot |\cos\phi|)^2 + (\tau_w \cdot |\sin\phi|)^2} \qquad (1)$$

where $\phi$ is the angle between wave and current direction and $\tau_m$ is defined as follows:

$$\tau_m = \tau_c \cdot \left[1 + 1.2 \cdot \left(\frac{\tau_w}{\tau_w + \tau_c}\right)^{3.2}\right] \qquad (2)$$

To include the effect of less frequent high waves for sediment mobilization and transport, the wave-induced shear stress generated by the highest waves of the given spectrum

$$H_{max} = 1.86 \cdot H_s \qquad (3)$$

estimated after EAK (2002) was calculated likewise.



### 3.3 Sedimentological properties

Surface sediment samples taken at the investigated sites prior to deployment of the lander were analyzed in a laser diffrac-tometer (*Beckman Coulter LS$^{TM}$ 13 320*) to obtain grain size distributions. The critical shear stress $\tau_{crit}$ of the sediments was computed from the median grain size $d_{50}$ following Soulsby (1997). An earlier study shows the validity of this approach for

the assessment of morphodynamic processes (Krämer and Winter, 2016).

### 3.4 Micro-bathymetry scanning

The micro-bathymetry was recorded at roughly hourly intervals using rectified images of a laser line projected on the seafloor. The vertical accuracy of this method is in the millimeter range (Ahmerkamp et al., 2017). Laser and camera were mounted on a moving sledge with a horizontal traveling distance of 0.55 m. From the scattered point data obtained by the laser scans,

DEMs of the bathymetry with an area of ca. 0.35 m × 0.55 m and a spatial resolution of $\Delta x = \Delta y = 2.5$ mm were generated using the `xyz2grid` function from the *Generic Mapping Tools* program suite (GMT V5.2.1; Wessel et al. (2013)) (Fig. 2) with linear interpolation and equal weighting.

The bathymetry scans take around 5 minutes to complete. In relation to the morphodynamic timescale, individual scans are regarded as synoptic. The volume of relocated sediment was computed as the difference between two DEMs $\mathbf{Z}_{i-1}$ and

$\mathbf{Z}_i$ recorded at successive times $t_{i-1}$ and $t_i$. Assuming that all material is relocated within the observation area, the resulting volume is divided by two.

$$\Delta V = 0.5 \cdot \Sigma |\mathbf{Z}_i - \mathbf{Z}_{i-1}| \cdot \Delta x \cdot \Delta y \tag{4}$$

The volumetric reworking rates can be computed as

$$R = \frac{\Delta V}{A \cdot \Delta t} \qquad \left[\frac{m}{s}\right] \tag{5}$$

Where $\Delta t = t_i - t_{i-1}$ is the time between the scans and $A$ is the base area of valid grid cells of the DEM. The reworking rate is computed as volume per area and timespan in the units $[\mathrm{m^3 m^{-2} s^{-1}}]$ which result in an apparent velocity in the units $[\mathrm{ms^{-1}}]$. Both physical and biogenic reworking occur for only a few hours depending on supercritical shear stresses for physical reworking and temporal pattern of benthic fauna activity for biogenic reworking. For the sake of comprehensible numbers and because it is common in the relevant literature, the reworking rates were converted to $[\mathrm{mm\ d^{-1}}]$. The absolute volume of

relocated sediment per time interval can be computed by integrating the rates over the respective period.

### 3.5 Classification of reworking mode

The reworking mode was classified by visual inspection of individual difference DEMs. Physically driven ripple migration results in characteristic regular pattern of erosion on the stoss side and accretion on the lee side of the ripples (Fig. 3p–t). Biogenic reworking, in contrast, manifests in small irregular patches (Fig. 3v–z). In some occasions, the form of large

organisms like sea stars or flat fish can be detected in the difference DEMs (Fig. 3a–m). For this study, the respective areas





were masked as they do not contribute to sediment relocation. The reworking mode for a given time step was classified into either physically or biologically dominated. Due to the difficulty to spatially separate simultaneous biogenic and physical reworking observed, mixed reworking in a single scan was not evaluated.

During conditions with combined wave and current shear stress well below the threshold of motion, sediment may neverthe-
less relocate due to avalanching of critical bedform slopes or due to single short-term turbulent events (Amirshahi et al., 2018). Wherever no characteristic pattern of biogenic reworking was observed in the difference DEMs, the resulting reworking was classified as physical (see e.g., Fig. 5).

## 3.6 Bedform height

Ripple heights were determined using a statistical estimate:

$$\eta_s = 2 \cdot \sqrt{2} \cdot \sigma(\mathbf{Z}) \qquad (6)$$

Where $\sigma(\mathbf{Z})$ is the standard deviation of the elevation matrix $\mathbf{Z}$. This formulation is strictly only valid for 2D sinusoidal surfaces. Absolute heights in a field of 3D ripples may be overestimated by around 40% (Krämer and Winter, 2016). Nevertheless, this method can provide a stable estimate of relief variability and temporal variations thereof. Changes in ripple height from successive scans were used to compute ripple growth and decay rates.

## 15  3.7   Identification of benthic species and bioturbation potential

Multicorer (MUC) samples (n≥3) were taken at all stations prior to the deployment of the lander. The retrieved cores were divided into 5 cm horizons and sieved to extract the infauna. After the census and classification of the species, their individual bioturbation potentials and reworking modes were determined. Community bioturbation potential was determined following Queirós et al. (2013). Pattern of biogenic reworking were compared to descriptions of burrow structures and dimensions to
identify the key bioturbators for the observed changes.

## 3.8   Classification of physical forcing

The physical forcing situation can be classified by comparing the combined wave and current shear stress to the critical shear stress required for the mobilization of sediment (Fig. 4–8). Sub-threshold conditions exist when neither wave nor current induced nor combined maximum shear stresses exceed the critical threshold for sediment mobilization. Current dominated
forcing exists when current shear stresses exceed the critical transport threshold during peak tidal flow and waves motion does not reach down to the seafloor. For these situations, more or less well defined slack water periods with weak or no flow pertain around the turn of the tides and sub-threshold conditions persist for several hours. Combined current and wave forcing can be observed during storm events when waves effects reach the seabed.





# 4 Results

## 4.1 Exemplary situations of physical and biogenic reworking

A close observation of the micro-bathymetry under sub-threshold conditions for sediment mobilization shows irregular patterns of sediment erosion and deposition (Fig. 2, Fig. 3). They manifest in small depressions and mounds and a general roughening

of previously smooth sediment surface. These patterns are evidence for biogenic reworking of seafloor sediments by benthic organisms during their burrowing and foraging activities. Biogenic reworking patterns were observed for 7 out of 15 deployments. The physical and biogenic reworking patterns and resulting rates are variable depending on the forcing situation and interaction between forcing and fauna activity. To outline the range of reworking rates under different forcing conditions, five characteristic examples are described below. The results from all evaluated deployments are summarized in Tab. 1.

### 4.1.1 Shallow station: CCP-G (HE432, 09/2014)

Station CCP-G is located in an area of subaqueous dune fields southeast of Helgoland island. With dune heights of around 2 m and lengths of around 50 m, the bathymetry is locally very heterogeneous. This station is dominated by strong tidal forcing.

Current shear stresses were supercritical for a period of around 3 hours during flood (Fig. 4a). Wave action did not reach the seabed. After a period of bedform migration, high biogenic reworking activity can be observed. The average biogenic

reworking rate measured here amounts to 5.7 mm d$^{-1}$ which is 14% of the physical reworking rate yielding 41.7 mm d$^{-1}$. The maximum instantaneous biogenic rate even reaches 10.1 mm d$^{-1}$.

The characteristic pattern of local erosion and deposition (Fig. 3u–z) have a typical diameter of 15–20 mm. In combination with the census of benthic species acquired from MUC samples, the burrowing mud shrimp (*Callianassa subteranea*) was identified as the most likely initiator. Assuming a funnel-shaped structure with angle-of-repose slope for the upper part of the

burrow, the typical diameter agrees well with an average tube diameter of 6 mm for adult individuals of *C. subteranea* (Forster and Graf, 1995).

### 4.1.2 Intermediate station: NOAH-E

Station NOAH-E is located on the former northeastern bank of the Elbe palaeovalley in a water depth of 30 m in a fine sand setting ($d_{50} =$242 $\mu$m). The magnitude of tidal flow velocity and current-induced shear stress vary over the spring-neap cycle.

Waves are only effective for sediment mobilization during storm events.

The deployment during cruise HE417 (03/2014) was characterized by strong tidal flow in combination with wave action (Fig. 5). This led to supercritical conditions for sediment transport throughout the first six of eight hours of observation. Even during sub-threshold conditions, biogenic reworking patterns could not be observed. The average physical reworking rate was 44.1 mm d$^{-1}$.

Under neap conditions and without wave action during cruise HE432 (09/2014), the critical shear stress was not exceeded for the entire observation period (Fig. 6). The changes observed in the micro-bathymetry were therefore entirely attributed to





biological activity. Locomotion of large animals like star fish and flat fish could be observed in the micro-bathymetry scans. Affected areas were manually masked since they generate a large apparent signal but did not correspond to actual sediment reworking. The average biogenic reworking rate amounted to $2.9 \, \mathrm{mm \, d^{-1}}$ and increased to a maximum value of $4.7 \, \mathrm{mm \, d^{-1}}$ during three single events which exhibit characteristic erosion-deposition patterns (Fig. 3b, k, m).

During cruise HE447 (06/2015), current shear stresses were supercritical but wave action did not reach the seabed (Fig. 7). The average physical reworking rate amounted to $28.3 \, \mathrm{mm \, d^{-1}}$. Biogenic activity identified during sub-threshold conditions amounted to an average reworking rate of $3.6 \, \mathrm{mm \, d^{-1}}$. The reworking pattern suggest involvement of a member of *Ophiuroidea*, found also in the corresponding MUC samples.

### 4.1.3    Deep station: CCP-J (HE432, 09/2014)

Located southeast of the Dogger Bank, station CCP-J lies in a water depth of 48 m. The seafloor sediment consists of fine sand ($d_{50} = 131 \, \mu\mathrm{m}$) with a mud content of 5–10% (Figge, 1981). As the sediment is not fully consolidated, the lander settled over the first five hours of the deployment (Fig. 8). Apparent reworking rates from this period could not be used. Wave action could be neglected and current shear stresses did not exceed the critical threshold. The observed biogenic reworking pattern resulted in an average rate of $1.5 \, \mathrm{mm \, d^{-1}}$.

**4.2    Identification of bioturbators**

The observed bioturbation pattern cannot be related to any of the three key bioturbators in the German EEZ, *Amphiura filiformis*, *Echinocardum cordatum* and *Nucula nitidosa*, identified by Wrede et al. (2017). However, none of said species has an high individual contribution to bioturbation potential in the investigated areas (compare Wrede et al., 2017, Fig. 2B–D). Due to the small area covered by the scans, the contribution of relatively large organisms is likely underestimated if they are not

present in the scan area by chance. The method presented here rather provides an estimation of the cumulative background reworking activity provided by smaller species than by one or few large species.

### 4.3    Bedform decay and form roughness

A significant reduction of bedform heights due to biogenic sediment reworking was observed only at station CCP-G during cruise HE432 (09/2014). Here, the bedform height is reduced by around 10% (from 2.4 to 2.2 cm) in a matter of 3 hours during

the high water slack period (Fig. 4b). Over the previous flood, bedform heights had increased from a similar level, indicating a dynamic equilibrium between physical growth and biogenic decay.

     The reduction of bedform heights results in a reduction of form roughness $k_{s,f}$ by 10% using the relation of van Rijn (1984) and a reduction of roughness height $z_{0,f}$ by 16% using the relation of Soulsby (1997).

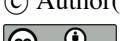



### 4.4 Relation of biogenic reworking rates to physical parameters

The observed biogenic reworking rates were compared to the physical parameters water depth $d$, median grain size $d_{50}$, bottom water temperature $T$ and current velocity $u$ (Fig. 9). Time-averaged values were used for the varying quantities. Including only observations with biogenic only or mixed reworking ($R_{bio} > 0$), the following linear relations can be expressed with the individual physical quantities. $R^2$ is the coefficient of determination or squared Pearson correlation coefficient for the linear regressions.

Water depth $d$ [m]:

$$R_{bio} = 8.415 - 0.151 \cdot d \quad \left[\text{mm d}^{-1}\right] \qquad R^2 = 0.754 \tag{7}$$

Median grain size $d_{50}$ [$\mu$m]:

$$R_{bio} = 1.061 + 0.008 \cdot d_{50} \quad \left[\text{mm d}^{-1}\right] \qquad R^2 = 0.788 \tag{8}$$

Bottom water temperature $T$ [°C]:

$$R_{bio} = -0.617 + 0.265 \cdot T \quad \left[\text{mm d}^{-1}\right] \qquad R^2 = 0.669 \tag{9}$$

Current velocity $u$ [ms$^{-1}$]:

$$R_{bio} = -0.296 + 32.786 \cdot u \quad \left[\text{mm d}^{-1}\right] \qquad R^2 = 0.673 \tag{10}$$

The linear regressions indicate positive correlations of biogenic reworking rates with median grain size, temperature and flow velocity and a negative correlation with water depth. The relations with water depth and median grain size do not explain the variability found between different observations at the same station and the fact that for some deployments, no biogenic reworking was observed. Temperature and flow velocity on the other hand, allow for the definition of thresholds for biogenic reworking activity. A lower threshold for bottom water temperatures of $T < 10$°C (Fig. 9c) separates the observations with no biogenic reworking with one exception (432B). Flow velocities $u$ exceeded $0.15$ ms$^{-1}$ for all observations without biogenic reworking (Fig. 9d).

Integrating the individual relations of the physical parameters with the biogenic reworking rate, the number $k$ is defined as follows:

$$k = \frac{T}{T_{opt}} \cdot u \cdot \frac{d_{50}}{d} \tag{11}$$

Where $T_{opt}$ is an assumed optimal temperature for benthic activity, here $T_{opt} = 25$° C. With three positive and one negative linear relations, the combined relation between the biogenic reworking rate and $k$ can be expressed in the form of a root function within the range of validity defined by the thresholds:

$$R_{bio} = 3.524 \cdot \sqrt{k} \quad \left[\text{mm d}^{-1}\right] \qquad R^2 = 0.759 \tag{12}$$

$$R_{bio} = 0 \qquad \text{If } T < 10°\text{C or } u > 0.15 \text{ ms}^{-1} \tag{13}$$



## 5   Discussion

### 5.1   Biogenic surface reworking and bioturbation potential

Recent studies highlight that the activity of benthic communities responsible for bioturbation are highly variable both tempo-
rally and spatially over several orders of magnitude: The temporal scales of variability in benthic fauna activity range from
semi-diurnal and diurnal cycles as a response to tides and daylight to seasonal cycles as a response to environmental parameters
such as water temperature and food availability (Oehler et al., 2015; Gogina et al., 2017; Morys et al., 2017). Spatial differences
between different habitats may vary mainly with abundance of active bioturbators or the general composition of the benthic
community. This makes a general assessment of bioturbation potential or even a quantification of biogenic reworking rates
very complex.

The observed changes in the small scale bathymetry between succesive laserscans and resulting volumetric reworking rates
do not show a correlation with bioturbation potential (BP) – neither community bioturbation potential ($BP_c$) nor cumulative
BP for selected species known as surficial reworkers. Therefore, it is important to note that the observed surficial changes
may cover only a part of the activity expressed in the concept of bioturbation potential. Nevertheless the direct measurement
of volumetric reworking rates is a promising approach to quantify the reworking activity and contribution of benthic fauna to
turnover of marine sediment and nutrients and pollutants entrapped in its pore space.

### 5.2   Physical boundary conditions for biogenic reworking activity

The linear relations given in Eq. 7–10 allow an approximation of biogenic reworking rates from readily available oceanographic
parameters. The correlations with water depth and median grain size most likely differentiate different benthic communities
(Reiss et al., 2009; Meyer et al., 2018) with different reworking effort. For the correlations between biogenic reworking and
the physical quantities temperature and flow velocity, the following causal relations are plausible.

#### 5.2.1   Temperature

The strong seasonal variability of reworking activity of individual species has been shown to correlate with environmental
conditions (Nichols, 1974; Cadée, 1976; Berkenbusch and Rowden, 1999). The relation between temperature and bioturbation
activity of dedicated species is documented by laboratory experiments (e.g., Braeckman et al., 2010).

The temperature $T_{opt} = 25°C$ chosen here lies beyond the upper range of the temperatures observed. Definitions of an
optimal temperature corresponding to a peak in reworking activity exist for individual species (Berkenbusch and Rowden,
1999; Ouellette et al., 2004). Yet, the obervations presented here do not indicate a peak in benthic reworking activity. This can
either mean that the optimal conditions are beyond the upper limit of the temperature range or that lower reworking activity
of one species is balanced by others with a different temperature optimum. The chosen value should therefore be carefully
tested against more observations. The concept also must not be extended to areas where higher temperatures induce hypoxic





or anoxic conditions (such as in the Baltic Sea), which would clearly lead to reduced abundance and activity of benthic fauna (Diaz et al., 1995; Modig and Olafsson, 1998; Conley et al., 2009).

### 5.2.2 Flow velocity

The upper threshold of $u > 0.15 \ \mathrm{ms}^{-1}$ for the observation of biogenic reworking may indicate that small benthic species may not find calm enough conditions at or near to the sediment surface for larger flow velocities. On the other hand, the activity of benthic fauna may simply not be observable with the given scan frequency of around one hour when biogenic reworking patterns are washed out by high bedload transport rates and fast ripple migration.

Two causal models for more biogenic reworking activity in response to higher sediment transport rates associated with faster flow velocities are possible.

1. Where currents generate transport conditions only for a few hours around peak tidal flow, intermittently bedload transport replenishes the food source for deposit feeders by entrapping fresh organic matter in the pore space of overturned sediment. Without previous transport events, the surface sediments may be rapidly depleted of food resources and thus become less attractive for deposit feeders and associated predators, reducing their reworking activity.

2. As a response to the overturning of the upper few centimeters of the seabed by migrating bedforms, burrowing suspension feeders are periodically forced to renew or clean out the upper part of their burrows to regain access to the water column.

### 5.3 Biogenic bedform decay

Because of the relatively small area covered by the laserscanner only a few ripples can be observed. Therefore, estimated bedform dimensions have to be interpreted carefully during phases of active ripple migration. Lee slope angles of mobile bedforms may exhibit local angles above the critical slope angle of the sediment because they are stabilized by flow reversal in the flow separation zone. When current shear stresses drop below the critical threshold for sediment transport, the slopes are left in an unstable state and easily avalanche towards the lower stable angle of repose and reduce bedform heights over the sub-threshold phase without biological activity.

The declining bedform height observed during HE432 at station CCP-G (Fig. 4b) with visual evidence for bioturbation highlights that the decay rates of ripple height due to the redistribution of sediment by benthic organisms may reach up to the order of 10% of the height within a few hours. Physically driven initialization, growth and decay of wave and current ripples are relatively well understood and included in semi-empirical models. Although formulations exist for biogenic bedform decay (Soulsby et al., 2012), absolute rates cannot be specified without field observations. The process of biogenic reworking is influenced by many factors and acts on various spatial and temporal scales; the decay rate observed here may serve as a first estimate of the order of magnitude of biogenic ripple decay by bioturbation.

This decay rate would flatten out a rippled seabed of 2 cm height within two to three days. However, current ripples in tidally dominated environments become periodically active and after a reduction quickly grow towards their equilibrium height (Bartholdy et al., 2015). The tidal forcing therefore interrupts biogenic decay and a dynamic equilibrium between alternating



periods of physically driven ripple growth and biogenic decay may be reached. Given enough time under sub-threshold conditions for transport, e.g. under neap tide conditions, one might expect to see a complete flattening of a rippled topography. However, this cannot be observed at station NOAH-E during cruise HE432 (Fig. 6). This may be explained by a different benthic community composition or reworking mode less effective for bedform decomposition.

## 6   Conclusions

The in situ observation of sediment reworking by benthic fauna presented here provides a first quantitative estimate of the biogenic contribution to sediment overturning under natural conditions and shows its relevance in comparison to physically driven sediment turnover. The average biogenic reworking rates at six different stations in the southeastern North Sea recorded over different seasons lie in a relatively narrow range of 1.5–5.7 mm d$^{-1}$. This is an order of magnitude less than the average

physical reworking rates. Biogenic reworking was found to be an omnipresent process limited only by either bottom water temperatures $T < 10°$ C or flow velocities $u > 0.15$ ms$^{-1}$.

The high biogenic reworking rates observed at station CCP-G (HE432), amounting to 14% of the average physical reworking rate, show the important contribution of the benthic community to sediment reworking even under energetic physical forcing. While physical reworking only occurs around peak tidal flow, the benthic organisms continue to overturn sediment during the

hydrodynamically calm phases. As the observations under neap conditions at station NOAH-E during cruise HE432 show, biogenic reworking may at times even be the dominant mechanism for sediment reworking when sediment transport conditions are only reached during spring tide. For deeper areas with little current forcing (e.g., HE447, NOAH-F), which are only affected by high wave action during seasonal storm events, the role of bioturbators becomes even more important.

By overturning sediment under hydrodynamically inactive conditions, the benthic fauna sustains cycling processes at the

benthic interface and provides an important regulating ecosystem service. Due to its reworking activity, the upper sediment pore space is flushed with oxygen rich water and fresh organic material from the overlying water column. This continuous cleansing of the pore space makes sandy sediments more effective for the filtration of nutrients (Ahmerkamp et al., 2015; Marchant et al., 2016).

By comparing the critical shear stress of the bed sediment with the actual sustained wave- and current shear stress, areas and

periods can be identified where or when bioturbation locally or temporarily dominates sediment reworking. The relations with physical boundary conditions, especially temperature and flow velocity, allow for an approximation of the average biogenic reworking rates from readily available oceanographic parameters (Eq. 12). While the correlations still have to be tested against a larger dataset, they allow for an extrapolation and assessment of the biogenic reworking effort to larger areas and longer periods.

*Data availability.* Relevant data will be made available on PANGAEA (https://pangaea.de/).




*Author contributions.* C.W. (HE417, HE441, HE447, HE470) and M.H. (HE432) planned and lead the cruises and conceived the sampling strategies. M.H. and S.A. operated the *Lance* lander. S.A. processed raw laserscan data. U.S. collected benthic fauna samples, identified species and calculated bioturbation potentials. K.K. analyzed laserscan and hydrodynamic data. K.K. conceived and wrote the manuscript in close discussion with C.W. and with input from M.H., S.A. and U.S..

5   *Competing interests.* The authors declare no competing interests.

*Acknowledgements.* The study was funded by the DFG-Research Center/Cluster of Excellence *The Ocean in the Earth System* at the University of Bremen (https://www.marum.de), the BMBF project *NOAH* (North Sea - Observation and Assessment of Habitats, Grant numbers: 03F0742A-E, 03F0743A, 03F0744A), the Max-Planck-Gesellschaft (https://www.mpg.de), and the Helmholtz Research Programme PACES II Topic 2. The authors are thankful for the professional support of captain and crew of R/V *Heincke* (Grant numbers: HE417, HE432, HE441, HE447, HE470) and a good working atmosphere during the cruises. A large part of the data analysis and figure layout for this manuscript was performed with the help of free and open source software. K.K. thanks the developers and community behind GMT, gdal, QGIS and Inkscape.



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



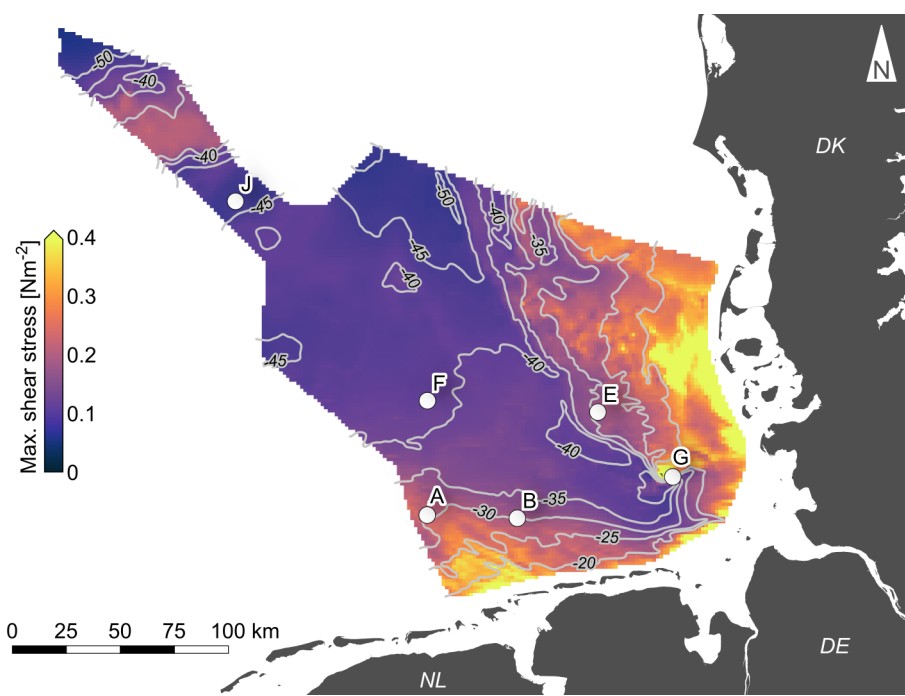

**Figure 1.** Distribution of the maximum shear stress as the combination of wave and current shear stress (Eq. 1) in the German EEZ averaged over the year 2014. Letters A–J denote the locations of repeated lander deployments. Shear stress data were retrieved from the coastMap Geoportal (www.coastmap.org) (Helmholtz-Zentrum Geesthacht Zentrum für Material- und Küstenforschung GmbH (HZG), 2017) (CC BY-NC 4.0). Bathymetry data were made available by the project Geopotential Deutsche Nordsee (GPDN) (2013). Land polygons ©OpenStreetMap (2016) (Open Database License).





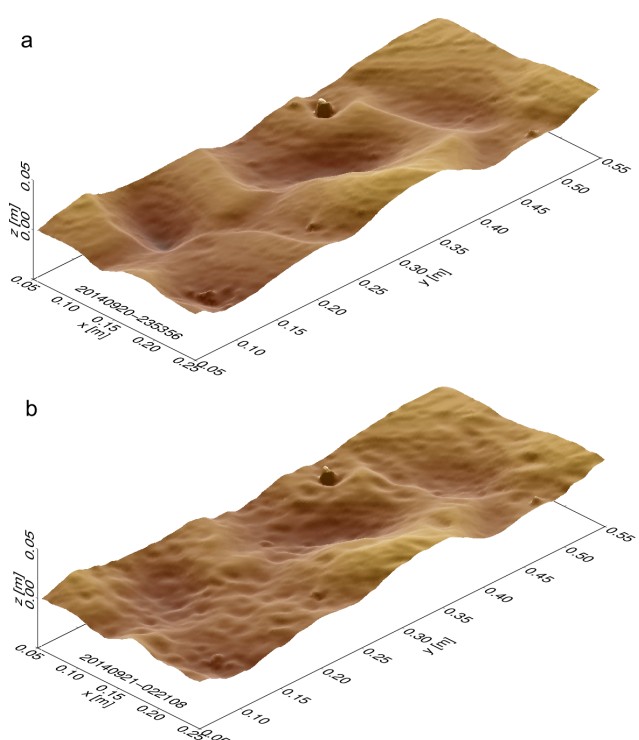

**Figure 2.** False-color DEMs of micro-bathymetry scans at station CCP-G recorded during cruise HE432, 20–21 September, 2014. (a) Mobile bedforms with smooth slopes at 23:53 local time. (b) Surface roughened by bioturbation at 02:21 local time.


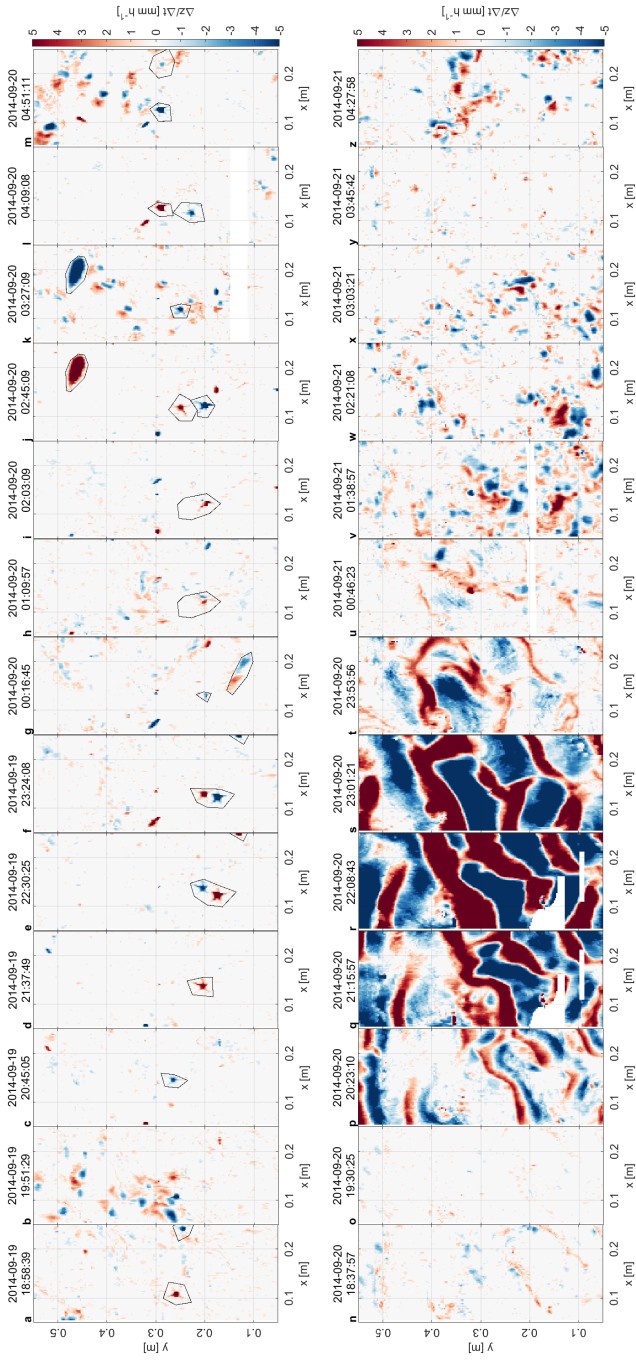

**Figure 3.** Difference DEMs for deployment at station NOAH-E (a–m) and station CCP-G (n–z) during cruise HE432. Black polygons outline manually masked areas affected by locomotion of large fauna (e.g. *Asteria rubens* (a, c–f, j–m) or *Pleuronectes platessa* (j, k)) without a contribution to sediment relocation. Characteristic pattern of physical reworking by ripple migration (p–t) and biogenic reworking (v–z) can be observed.

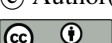


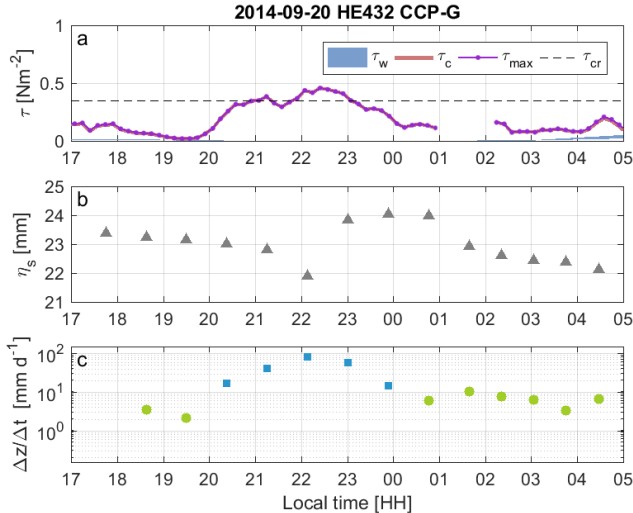

**Figure 4.** Deployment at station CCP-G during cruise HE432. (a) Wave, current and maximum combined shear stress versus critical shear stress. As wave forcing was mostly irrelevant, the maximum combined shears stress is mostly constituted by the current shear stress. (b) Ripple height. (c) Physical (blue squares) and biogenic (green circles) reworking rates.

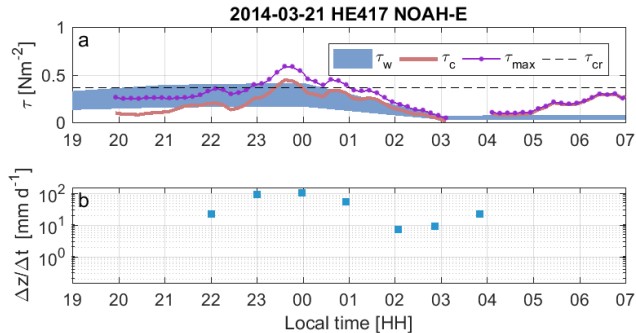

**Figure 5.** Deployment at station NOAH-E during cruise HE417. (a) Wave, current and maximum combined shear stress versus critical shear stress. (b) Reworking rates.



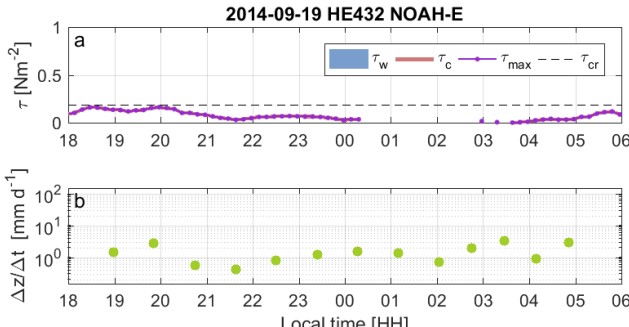

**Figure 6.** Deployment at station NOAH-E during cruise HE432. (a) Wave, current and maximum combined shear stress versus critical shear stress. As wave forcing was not relevant, the maximum combined shears stress corresponds to the current shear stress. (b) Reworking rates.

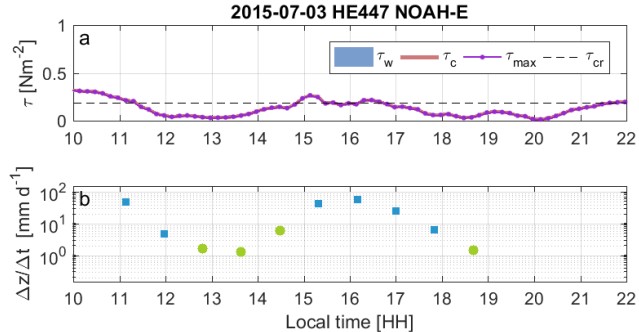

**Figure 7.** Deployment at station NOAH-E during cruise HE447. (a) Wave, current and maximum combined shear stress versus critical shear stress. As wave forcing was not relevant, the maximum combined shears stress corresponds to the current shear stress. (b) Physical (blue squares) and biogenic (green circles) reworking rates.

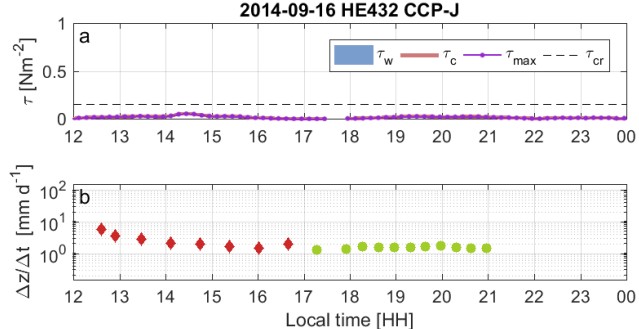

**Figure 8.** Deployment at station CCP-J during cruise HE432. (a) Wave, current and maximum combined shear stress versus critical shear stress. As wave forcing was not relevant, the maximum combined shears stress corresponds to the current shear stress. (b) Biogenic reworking rates (green circles). Red diamonds indicate invalid data due to the settling of the lander into the unconsolidated sediment during the first five hours of the deployment.



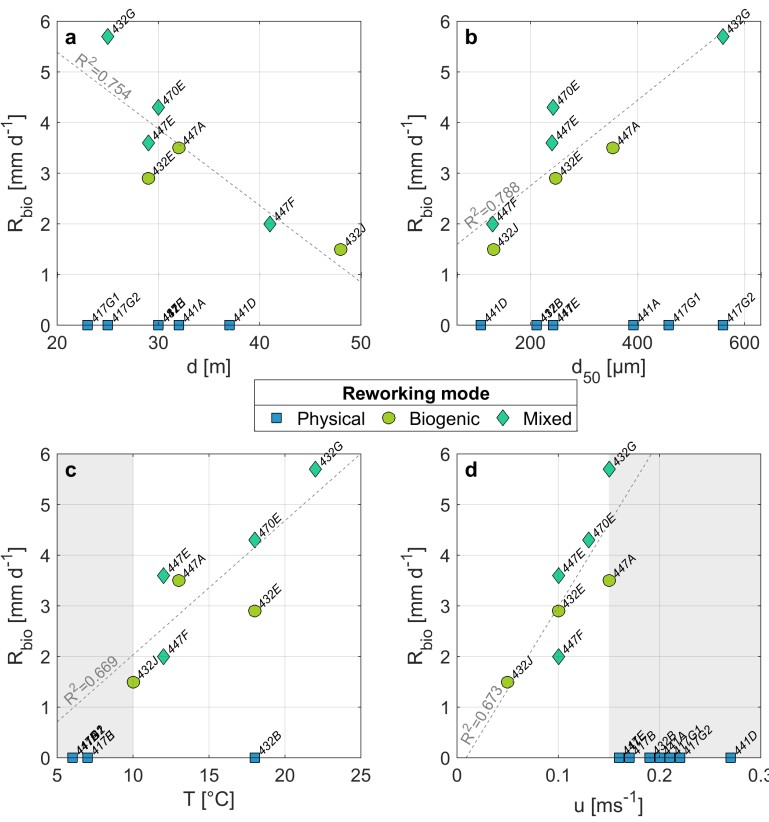

**Figure 9.** Biogenic reworking rates in relation to physical parameters. (a) Water depth, (b) median grain size, (c) bottom water temperature and (d) current velocity. The gray areas in (c) and (d) mark conditions under which no biogenic reworking was observed (Eq. 13).





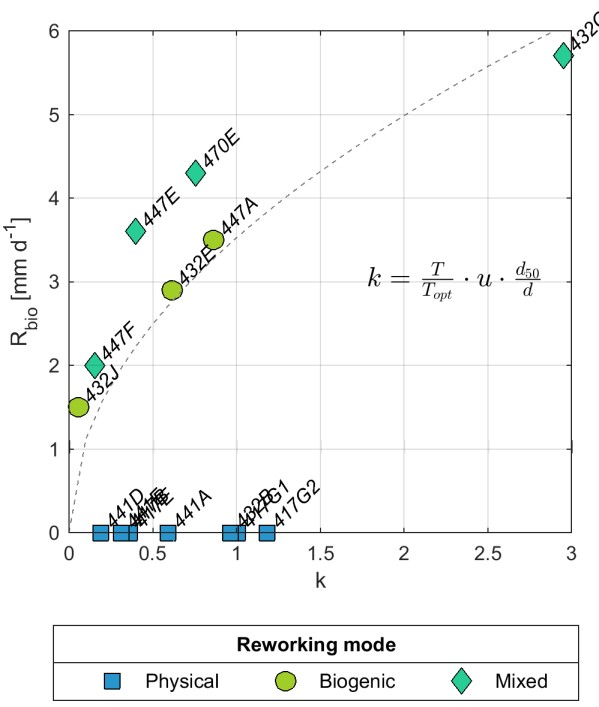

**Figure 10.** Relation between biogenic reworking rate and the combination of water depth, median grain size, bottom water temperature and current velocity. The dashed line denotes the approximation by Eq. 12.





**Table 1.** Deployment meta data and physical and biogenic reworking rates obtained from difference DEMs.

| Cruise/Station | Deployment | | Position | | Environment | | | | | Reworking rate | |
| | Date | Duration | Lat | Lon | Depth $d$ | Temp. $T$ | Velocity $u$ | Grain size $d_{50}$ | For-cing | Physical $R_{phy}$ | Biogenic $R_{bio}$ |
| | | [hh:mm] | [°] | [°] | [m] | [°C] | [ms$^{-1}$] | [$\mu$m] | | [mm d$^{-1}$] | [mm d$^{-1}$] |
| 417B | 2014-03-13 | 03:39 | 53.98650 | 6.86783 | 30 | 7 | 0.17 | 211 | c | 9.5± 2.3 (13.50) | 0 |
| 417G1 | 2014-03-17 | 09:31 | 54.16850 | 7.98850 | 23 | 6 | 0.21 | 458 | cw | 85.8±13.0 (106.30) | 0 |
| 417G2 | 2014-03-20 | 07:30 | 54.17317 | 7.95867 | 25 | 6 | 0.22 | 559 | cw | 56.6±44.3 (109.70) | 0 |
| 417E | 2014-03-21 | 08:06 | 54.44433 | 7.41450 | 30 | 7 | 0.16 | 242 | cw | 44.1±39.8 (103.00) | 0 |
| 432J | 2014-09-16 | 08:30 | 55.25833 | 4.74967 | 48 | 10 | 0.05 | 131 | s | 0 | 1.5±0.1 (1.7) |
| 432E | 2014-09-19 | 10:45 | 54.43750 | 7.42267 | 29 | 18 | 0.10 | 247 | s | 0 | 2.9±1.0 (4.7) |
| 432G | 2014-09-20 | 10:42 | 54.17317 | 7.95867 | 25 | 22 | 0.15 | 230 | c | 41.7±27.9 (80.4) | 5.7±2.6 (10.1) |
| 432B | 2014-09-23 | 08:29 | 53.98717 | 6.87050 | 30 | 18 | 0.19 | 211 | cw | 54.6±37.2 (110.5) | 0 |
| 441D | 2015-03-20 | 02:13 | 54.09250 | 7.35850 | 37 | 6 | 0.27 | 107 | cw | 52.0±22.5 (80.4) | 0 |
| 441E | 2015-03-24 | 05:56 | 54.44133 | 7.41250 | 30 | 6 | 0.16 | 242 | cw | 51.0±45.7 (110.1) | 0 |
| 441A | 2015-03-26 | 09:01 | 53.98717 | 6.22483 | 32 | 6 | 0.20 | 392 | cw | 10.7± 6.0 (24.8) | 0 |
| 447A | 2015-06-28 | 08:18 | 53.98850 | 6.23083 | 32 | 13 | 0.15 | 354 | s | 0 | 3.5±1.0 (5.3) |
| 447F | 2015-06-29 | 12:20 | 54.46850 | 6.19300 | 41 | 12 | 0.10 | 129 | c | 10.5± 7.9 (24.3) | 2.0±1.4 (4.2) |
| 447E | 2015-07-03 | 08:39 | 54.43883 | 7.42300 | 29 | 12 | 0.10 | 240 | c | 28.3±20.9 (50.6) | 3.6±1.7 (6.0) |
| 470E | 2016-08-28 | 11:50 | 54.44117 | 7.41817 | 30 | 18 | 0.13 | 242 | c | 4.7± 1.0 (5.6) | 4.3±1.5 (7.4) |

Reworking rates are given as: $avg \pm std.\ (max.)$. The forcing classification indicates supercritical shears stress due to current (c), waves (w) or both (cw) and sub-threshold conditions (s).