# Peer review of "Relations of physical and biogenic reworking of sandy sediments in the southeastern North Sea"

_Ocean Science, 2018_

## Referee Comment (RC1) · Anonymous Referee #1 · 7 Mar 2019

Review of 'Relations of physical and biogenic reworking of sandy sediments in the southeastern North Sea' by Krämer et al.

Summary.

The manuscript describes a set of observations from a number of sites in the German Bight, including sediment characteristics, current velocities and sea-bed micro topography. Successive scans of the sea-bed micro topography are subtracted to reveal changes in bed level, which are interpreted in terms of physical and biological reworking of the sea bed. Linear relationships are fitted through the biological reworking data as a function of environmental characteristics.

General comments.

[Figure]

The topic of the manuscript is interesting and relevant. The manuscript reads well, and I found very few grammatical or typographical errors. The figures are clear and easy to read. The primary data set is interesting and relevant, and worth publishing. However, the structure can be improved (Data and Methods contains elements of Results and vice versa). Also, lots of detail is missing, to the level that it would be next to impossible to replicate the work/results. Moreover, I have fundamental concerns regarding the analysis and interpretation. I will explain all of this below. Overall, I think this takes the manuscript beyond major revisions (which would come with a deadline that's too tight to do the additional work required to bring it to a level that is publishable). As the basic data are sound and interesting, and the authors write well, I would suggest rejection, and encourage the authors to re-think the analysis, re-write the manuscript, and re-submit when they are ready.

Analysis - main concerns.

1. The authors use changes in surface level as a measure for reworking. For migrating bed forms, if done at a high enough temporal frequency, this gives a good estimate of the volume of bed material that is moved. However, this does not hold for biogenic reworking by burrowing organisms. These may move a whole column of sediment down to their maximum burrowing depth, with only minimal changes in sea-bed elevation. Hence, at best, the results presented for biogenic reworking represent a lower boundary for the range of potential true values. As a result, the terminology (up to the title!) is misleading, and 'surface level change' (or something equivalent) should be used instead of 'reworking'. Also, these caveats should be stated clearly.

2. The current manuscript also does not use the species analysis from the box cores to its full potential.

3. The main analysis, described in Section 4.4, is unclear, seemingly constructed from random bits, and the result is demonstrably wrong. Why use 'time-averaged values for the varying quantities' when the reworking rates are instantaneous values (eqn 5) -

or are they? If the reworking rates are also time-averaged, why was this done before regression? Surely regression can also be carried out (and better) on instantaneous data? How was the time averaging done - in the same way for all the quantities? Over an exact tidal cycle so there are no truncation errors? Or different for the biogenic reworking which is only active part of the time? Why have separate linear regressions if a function (eqn 11, 12) is available (I now guess that it may be a step in constructing 11 and 12, but this is not clear from the text - and this doesn't make it a correct approach)? Why linear - are the processes expected to be linear? Apparently not (eqn 11, 12). Why is there a mis-match in units (meters, milli-meters, micro-meters)? Consistent units (m, s) should be used throughout. Where do eqns 11 and 12 come from? Why would burrowing organisms respond like this? How can eqns 11,12 be correct (or an un-biased phenomenological relationship) if all but one of the data points are above the functional curve (Fig 12)? Another argument why it can't be correct is that the 'constant' 3.524 has units (m^0.5 s^-0.5 - ignoring the mess of m vs mm, sec vs day for simplicity), and hence contains part of the processes. A proper phenomenological function with should have fitted constants that are non-dimensional. What, in the end, is the physical/biological meaning of eqn 11,12? Why use current velocity, and not current shear stress and wave shear stress which were both shown to be important earlier on? This should all be re-done, using a uniform set of units, starting from relationships that make biological/physical sense, and using dimension analysis to plug the gaps, and using multi-variate regression if/where appropriate. I am not sure if this kind of approach is realistic and feasible, also given the relatively few data points that the authors have. One option they could consider is to abandon this approach and do something more feasible with the data?

Structure.

1. To improve the flow, Section 3.3 should come after Section 3.6.

2. Figures 2 and 3 are presented in Data and Methods, but they are Results.

3. There needs to be a section before the current 4.1 that describes the results of the core samples in terms of species: numbers and biomass. In 2.4, these species need to be described in terms of size and bioturbation characteristics (type, depth and activity), incl. references. These data need to be included in the discussion/interpretation, including in the new analysis (4.4).

4. Section 4.4 is largely Theory/Methods - split the derivation of the methodology to a section in 3 and keep the implementation/numbers/results in 4.

5. Figure 10 is not referenced nor discussed.

Detail.

Title: needs changing to reflect definition of 'reworking' - see above.

Results: past tense should be used throughout to describe the results.

p. 2, l. 21 patterns of

p. 3, l. 12. patterns of ... with water depth

p. 3, l. 16-17. Requires a reference. Tidal currents are not necessarily small near amphidromic points (quite the reverse!)

p. 3, l. 18. regularly observed for

p. 3, l. 20. semi-monthly time scale

p. 3, l. 20. adding: to what?

p. 3, l. 20,22. 'subcritical' vs 'sub-threshold': please use uniform terminology throughout. I have reservations about 'sub/supercritical' as these terms have a different meaning for fluids/gasses in thermal physics.

p. 3, l. 24-29. This paragraph requires references. Also please discuss presence of megaripples and sand waves.

p. 3, l. 25. migrate over a distance of the order

p. 4, l. 1. There are five macrofauna communities in the southeastern North Sea.

p. 4, l. 1-6. It would help the non-biologist if common names are also given. Moreover, are these communities (as said) or species? If communities, surely they consist of multiple species - please describe. Please also include here a description of the species found and analysed in the box cores, including their burrowing characteristics (type, depth, activity, timing of activity, etc.).

p. 4, l. 9-14. Please include a diagram of the lander configuration with instruments.

p. 4, l. 14. ...12 hours after which the lander was recovered. (?)

p. 4, l. 15. 10-minute averaged: why?

p. 4. Refs to Soulsby. Please remove page numbers - this does not conform with the journal standard. Also they are apparently not consistent between prints (they are different in my copy).

p. 4, l. 17. Only data with...: how frequently did this occur, i.e. how much data was discarded?

p. 4, l. 18-19. Waves from model. This surprises me. Can this not be derived from the ADCP data?

p. 4, l. 25-28. Soulsby used field data to fit their model, so the influence of maximum waves is already included in eqs 1 and 2. Remove this, and remove it from all the calculations.

p. 5, l. 3. laser diffractometer. Please provide resolution and accuracy.

p. 5, l. 5. tau_crit, d50. Please provide equations. How was d50 derived from the size distributions?

p. 5, l. 7. roughly. How roughly? Why not exactly? Why not more often? How does

this affect the calculations (eqn 4,5)?

p. 5, l. 9. sledge. This suggests something pulled along the sea bed, which I don't think is the case? Is there a better word for this? Carriage? See also remark about adding a diagram of the lander.

p. 5, eqn 4 and 5. It would be better to use central differences in time rather than upwind? The current formulation results in an R at t(i-1/2). How are the other data treated on this time frame?

p. 5, l. 23. and the temporal pattern

p. 5, l. 28. in a characteristically regular pattern

p. 5, l. 29. manifests itself in

p. 6, l. 3. was not identified. (?)

p. 6, l. 7. Figure 5 is reference before figure 4.

p. 6, l. 6-7. I don't recognise what's described in the text in what's plotted in Fig 5.

p. 6, eqn 6. Needs a reference.

p. 6, par. 3.6. Is the sampled area big enough, i.e. does it contain enough ripples, to get unbiased estimates of ripple height?

p. 6, l. 17. species: which?

p. 6, l. 18 bioturbation potentials and reworking modes were determined. How? Also these are not in Results - please add.

p. 6, l. 19. descriptions of burrowing structures and dimensions: reference and provide these descriptions, how was the comparison done?

p. 6, l. 20. bioturbators responsible for the observed changes in bed level. (?)

p. 6, l. 22. critical shear stress: requires equation and reference.

p. 6, l. 23. Fig 4-8. This is Results.

p. 6, l. 24. why separate, not just combined?

p. 7, l. 5. evidence of

p. 7, l. 6/7. deployments (Table 1).

p. 7, l. 12. station: where: on the top or in the trough of a dune?

p. 7, l. 15. reworking rate (41.7 mm d-1)

p. 7, l. 17. The characteristic spatial pattern of biogenic erosion ...20 mm and a depth of ***

p. 7, l. 19. identified: how?

p. 7, l. 23. remove former

p. 8, l. 2. did not contribute to the actual

p. 8, l. 4 characteristic biogenic erosion

p. 8, l. 7 suggest(ed): Why? Which other members were also involved?

p. 8, l. 11. Figge, 1981. Why rely on 40-year old data for this? This can well have changed! You've done particle size analyses - use your own data!

p. 8, l. 9. settled: how much? Also: this suggests a higher mud content than 5-10%?

p. 8, l. 16. cannot: why not?

p. 8, Section 4.2. This section needs to be improved and expanded, using specific information from the core samples.

p. 8, l. 23-24. refer to fig4b in this sentence.

p. 8, l. 27-28. Why is this here? Why is it relevant? It is not used further on. If it is relevant, please give the equations (these are thick books), justify in the Introduction

and discuss in the discussion. Also, I don't understand why this is the only station with a significant reduction in ripple height, as the rates of bed-level change are similar as for other stations (Fig 5, 7).

p. 9, eqn 7-10, 12. Please provide error estimates of the regression coefficients.

p. 9, l. 19. T<10. How do you know it's not T<7?

p. 9, l. 20. Flow velocities: it's crucial to know how these were averaged.

p. 9, l. 25. Topt=25. Why? (this comes later, but that bit should be moved to Methods)

p. 10, l. 10-12. This statement is not backed up with data.

p. 10, l. 15. pollutants contained in

p. 10, l. 17-18. No, this gives 4 values for R_bio that will generally not be the same.

p. 11, l. 10-15. Or both?

p. 11, l. 27-29. Quite a statement to make based on 1 sample! Suggest to remove this line of reasoning from the manuscript.

p. 12, l. 3, fig 6: does not display ripple height.

p. 12, l. 3. different: than what?

Figures.

Figure 3: the colour scale saturates - please improve the plots such that maximum and minimum values are shown properly. What are the white bars (missing data?)? How are these dealt with in the calculation of R?

Figure 4-8: delta_z/delta_t: these have a sign, please represent. Currently the figs suggest that the bedforms keep growing.

Figure 9-12: The data (R_bio, d, d50, T, u) are said to be averages. Please provide error bars for all of these calculated from the scatter of the data that compose these

averages.

Figure 5-8: why not plot ripple height as in Fig 4?

Figure 9: I'm confused, as on p.6, l. 1-2 it was stated that mixed was not determined, yet here it is...

[Figure]

---

## Referee Comment (RC2) · Anonymous Referee #2 · 18 Mar 2019

This manuscript provides an interesting measurement of biological reworking of benthic sediments at six locations in the North Sea. The authors used DEM measurements with shear stress calculated from ADCP measurements to observe changes in the seabed morphology on very small scales. With shear stress below a critical erosion, these changes are attributed to biological reworking of sediments, and is quantified. These are interesting and novel measurements.

I found the explanation of physical processes and calculations to be over simplified and missing information. This analysis relies on all changes in the sediment mapped to be due to biogenic reworking when the bed shear stress is below a critical threshold, no discussion is present about the intermittency of sediment transport below a the critical shear stress. Equations demonstrating the physical methods as well as plots showing

these measurements would help to clarify the work done. I would also like to see plots of physical parameters: tides, waves (Hs, DPD) to put the short-term benthic lander measurements into an environmental context.

There is no discussion of error or uncertainty.

I found the manuscript disjointed - e.g. the biogenic reworking was compared to temperature, velocity, grain size, without any introduction to how these things influence reworking. The measurements are novel, but in this state the manuscript reads as a compilation of measurements instead of a strong scientific story.

Further line-by-line comments below:

pg 1, 8 - The measured biogenic reworking rates reach up to 14% of physically driven reworking via bedform migration.

Reword this sentence, I initially read it that the biogenic processes are driving bedform migration.

(!) 2, 17 - I don't like this....

pg3, 2 - What about the bed below 40m depth? Your measurements extend to 48m.

pg3, 5 - Morphological features at the deeper stations with weak bottom currents and beyond the reach of surface waves are of mostly of biogenic origin.

Is there a seasonality to the morphological features? I would expect that at least part of the year 40 m would not be below the depth where surface waves are felt.

pg3, 12-20 - I feel that this paragraph repeats some common assumptions about waves and wave-resuspension in shelf seas that require justification. Specifically, "Wave-induced shear stresses therefore usually do not reach supercritical conditions for sediment motion from late spring to early fall." Work in the much-deeper Celtic Sea by Thompson et al. (2017) showed persistent tidal resuspension, and of more importance here - showed wave orbital velocities at ∼100m depth even in August. If this work

hinges on waves not being important to sediment movement in 20-40m in the North Sea in summer, analyses need to be present to show that this assumption is reasonable. The very basic rule-of-thumb is that if you assume linear waves, the wave will be 'felt' at the depth of half of the wavelength of the surface wave. From Grant and Madsen (1986): "The surface wave velocity and pressure fields penetrate to the seabed only in water depths less than about half their wavelength (e.g. Madsen 1976). For example, a 12-s wave, generally in the swell band, will penetrate to the bottom in 112 m of water or shallower, whereas a 6-s wave, typical of the wind-sea band, will penetrate to only 28 m or shallower." You can use

pg3, 26 - The activity and migration rate of ripples is mainly controlled by the magnitude of tidal flow, varying over the spring-neap cycle. Although they may not often become the dominant driver in ripple generation, the stirring effect of waves facilitates ripple migration.

This needs justification. Waves and currents both form ripples, and I'm not convinced from your previous paragraph that waves have been systematically discounted.

pg3, 28 - Citation? What about the relationship between mud content and ripples? E.g. Lichtman et al. 2018, Baas et al. 2013, 2016, Malarkey et al. 2015.

pg5, 3 - What is the formula used? I can't find reference to a critical shear stress equation from Soulsby (1997) used in Kramer and Winter (2016). Is the sediment distribution unimodal? A bimodal distribution can give you a nonsense d50, and might alter how sediments are mobilized (e.g. McCarron et al. 2019). Here you reference critical shear stress at tau_crit, but in the figures it is tau_cr, make it consistent.

pg5, 29 - How do you know you have caught all the organisms in your calculations?

pg7, 6 - Why wasn't there biogenic reworking in the other eight deployments?

pg7, 15 and elsewhere where values of R_bio are reported - What are the error bounds on these measurements?

pg7, 25 - Again, a qualitative assessment of waves vs. tidal suspension is missing. In Figure 5 waves are relevant. What were the oceanic conditions there compared to Figure 6? I would like to see figures of tides and waves instead of just text saying, "neap tide" or "wave action."

pg8, 2 - How do you know you masked correctly? Again, some calculations of the error in these values and in the approach is necessary.

pg8, 10-14 - Why isn't the sediment consolidated if the shear stresses are low? I would think this would be representative of a region with constant resuspension.

section 4.4 and pg12, 25- Why these parameters? Why not species composition, mud content, some metric of food availability? Are the currents the tidal currents? What about wave-induced currents? Your data show tidally variable conditions, please justify tidally averages here. Your figure 3 showed a period where ripples were obviously created, some the time variability in these measurements seems important, especially in u.

Pg. 17, Figure 1 caption: "Letters A-J" suggests these are inclusive of all letters, but C, D, H, and I are not present/used here. Rephrase: "The six stations with letters A, B, E, F, G, and J..."

Figures 4 - 8: I see here that your calculations of wave stress are indeed low except for one figure. In the introduction you discount waves as relevant, but it would be more appropriate to include their effects and show here in the results that in most cases wave-induced bed shear stress is very low. On that note, is there bias in the conditions of these measurements? I would assume limitations with the ship and cruise schedule probably biased toward calmer conditions.

The highest value of tau looks to be around 0.6 N/m2 so the axis doesn't need to go to 1 N/m2 - it creates empty space when better resolution of the figure would be nice. I think the purple is over the red line in most cases, but this is an inference. Please

improve these plots so it is obvious what is being shown - i.e. it could be either the blue or red line that is hidden by the purple.

References

Baas, J.H., Davies, A.G., Malarkey, J., (2013). Bedform development in mixed sand-mud: the contrasting role of cohesive forces inflow and bed. Geomorphology 182, 19–39.

Baas, J. H., Best, J. L., and Peakall, J. (2016). Predicting bedforms and primary current stratification in cohesive mixtures of mud and sand. J. Geolo. Soc. 173, 12–45. doi: 10.1144/jgs2015-024

Grant, W.D., Madsen, O.S. (1986). The continental-shelf bottom boundary layer. Annual Review of Fluid Mechanics 18, 265–305. doi:10.1146/annurev.fl.18.010186.001405.

Lichtman, I. D., Baas, J. H., Amoudry, L. O., Thorne, P. D., Malarkey, J., Hope, J. A., et al. (2018). Bedform migration in a mixed sand and cohesive clay intertidal environment and implications for bed material transport predictions. Geomorphology 315, 17–32. doi: 10.1016/j.geomorph.2018.04.016

Malarkey, J., Baas, J.H., Hope, J.A., Aspden, R.J., Parsons, D.R., Peakall, J., Paterson, D.M., Schindler, R.J., Ye, L., Lichtman, I.D., Bass, S.J., Davies, A.G., Manning, A.J., Thorne,P.D., (2015). The pervasive role of biological cohesion in bedform development. Nat.Commun. 6, 6257.https://doi.org/10.1038/ncomms7257.

McCarron, C. J., Van Landeghem, K. J. J., Baas, J. H., Amoudry, L. O., & Malarkey, J. (2019). The hiding-exposure effect revisited: A method to calculate the mobility of bimodal sediment mixtures. Marine Geology, 410, 22–31. https://doi.org/10.1016/j.margeo.2018.12.001

Thompson, C.E.L., Williams, M.E., Amoudry, L.O., Hull, T., Reynolds, S., Panton, A., and Fones, G.R. (2017). Benthic controls of resuspension in UK

shelf seas: Implications for resuspension frequency. Continental Shelf Research doi:10.1016/j.csr.2017.12.005.

---

## Short Comment (SC1) · 18 Mar 2019

Dear anonymous Referee,

thank you for your detailed comments and the positive feedback regarding the value of the primary dataset. Thank you also for your suggestions for improving the structure and flow of the manuscript.

We address your concerns about a) the amount of detail (in the methods) and b) the main analysis below, and are positive that these can be handled by a review of the manuscript within a few weeks time. Some of the criticized issues seem to be based on wording, maybe a misunderstanding of what we intend to point out with this dataset and our analysis: In our view the main value of this manuscript lies in the detailed field

observations of biogenic (and physical) overturning of sediments under natural conditions. The word overturning now is used in a sense of bed surface change, which certainly is not similar to the amount of material moved inside and at the bed (=bioturbation). The overturning by physical processes can be described with the well known concept of critical and effective shear stress. The observed biogenic overturning on the other hand is a much more complex process, and has so far been understudied under field conditions. We evaluated time-averaged values as a more robust descriptor for typical biogenic overturning of the sediment surface than the instantaneous rates. The narrow range of observed values for different settings shows the precision of the method but also the validity of said average overturning rates as typical values for the entire region. The empirical regressions with the chosen set of physical boundary conditions are provided as possible descriptors for the observed range in average biogenic overturning rates. The high correlation coefficients justify this approach given the small number of observations available at this point. We did not mean to provide a complex model for the interactions between physical boundary conditions and the activity of the benthic fauna.

Knut Krämer

*Analysis - main concerns.*

*1. The authors use changes in surface level as a measure for reworking. For migrating bed forms, if done at a high enough temporal frequency, this gives a good estimate of the volume of bed material that is moved. However, this does not hold for biogenic reworking by burrowing organisms. These may move a whole column of sediment down to their maximum burrowing depth, with only minimal changes in sea-bed elevation. Hence, at best, the results presented for biogenic reworking represent a lower boundary for the range of potential true values. As a result, the terminology (up to the title!) is misleading, and 'surface level change' (or something equivalent) should be used instead of 'reworking'. Also, these caveats should be stated clearly.*

> We agree that the term 'reworking' must be used carefully and includes the activity of the benthic fauna in the sediment volume up to their maximum depth of activity. The term was chosen because it is common in the related literature and actually used to describe only surficial changes (e.g., Grant, 1983). We think that 'surface level changes' would not grasp the importance of the mechanism for the exchange between sediment and water column (p. 1, l. 3-4; p. 12, l. 19-23). Instead, we would propose the term 'overturning' (of surface sediment) as it may help to describe the volumetric changes which transport material across the benthic interface i.e., the sediment surface, which is registered by the method. These terminology problems will be taken up in the discussion. The caveats are already partly discussed (p. 10, l. 12-15), but this will be extended.

*2. The current manuscript also does not use the species analysis from the box cores to its full potential.*
> Multicorer (MUC) (p. 6, l. 16) cores were used. Due to the small area and volume of sediment covered by this method, it may be unsuitable for a representative description of the benthic fauna. Indeed, no correlations of the observed biogenic overturning rates with bioturbation potential (p. 10, l. 10-13) were found. Given this, we decided to remove all information gained from the core samples and speculations related to individual species or benthic communities and present the observed biogenic overturning rates with an assumption based on earlier studies in the area.

*3. The main analysis, described in Section 4.4, is unclear, seemingly constructed from random bits, and the result is demonstrably wrong.*
> Obviously we could not get our message through. We answer the individual issues below:

*Why use 'time-averaged values for the varying quantities' when the reworking rates are instantaneous values (eqn 5) - or are they? If the reworking rates are also time-averaged, why was this done before regression? Surely regression can also be carried out (and better) on instantaneous data?*

> The regressions were meant to provide a first estimate of biogenic sediment overturning, compared to basic physical values describing the overall setting at the locations rather than touching complex instantaneous interactions. Most of the physical parameters used in the regressions are (more or less) constant over the observed period ($d$, $T$, $d_{50}$). Mainly the flow velocity changes throughout the tide. Thus a point-to-point regression with instantaneous biogenic overturning rates is not considered useful, as the fauna is only active (or can be observed) under sub-threshold conditions for sediment transport. Nevertheless, We agree that as indicator the average flow velocity may be misleading and the maximum flow velocity or shear stress observed may better represent this physical boundary condition. This would also help to overcome the problems with truncation errors noted below.

*How was the time averaging done - in the same way for all the quantities? Over an exact tidal cycle so there are no truncation errors? Or different for the biogenic reworking which is only active part of the time?*

> The averaging was done for the respective periods of either physical or biogenic activity. The observation cover only part of one tidal cycle due to the limited battery capacity (see Tab. 1). The chosen unit of [mm/d] may be misleading because it suggests that the measured rates were extrapolated to longer periods which is not the case. This will be changed. Not knowing the exact reworking by individual species, we consider an average rate. The rates provided were meant to give an idea of the typical overturning activity for a given station and time. We think that the good correlation of the average rates with the physical boundary conditions justifies this approach.

*Why have separate linear regressions if a function (eqn 11, 12) is available (I now guess that it may be a step in constructing 11 and 12, but this is not clear from the text - and this doesn't make it a correct approach)? Why linear - are the processes expected to be linear? Apparently not (eqn 11, 12).*

> We provide a first observation – not a complete model of the relation of physical and biogenic overturning. The individual regressions show possible relations of the

observed biological activity with individual oceanographic conditions. Linear regressions are suggested as a simple first approach to correlate the biogenic activity with the physical parameters. More complex relations would only make sense if we already had a model for the behavior of the benthic fauna with regard to the physical boundary conditions. The function in eq. 12 was not available but constructed by 'summarizing' the individual relations (p. 9, l. 22-23; eq. 11). We will not include the separate equations in the next version to avoid confusion; and just provide the best fit model.

*Why is there a mis-match in units (meters, milli-meters, micro-meters)? Consistent units (m, s) should be used throughout.*
> The unit [mm/d] is common in describing biogenic reworking rates in the related literature and results in comprehensible values. For the physical parameters, the common units (e.g., $d_{50}$ [$\mu$m]) were maintained. From the confusion this generated with regard to the validity period of the measurements (see comment above) and in the regressions we agree that it is better to abandon simplicity for the sake of consistent (SI) units.

*Where do eqns 11 and 12 come from?*
> Equation 11 was constructed by 'summarizing' the effects of all individual relations (eq. 7-10) in the the factor
$k = \frac{T}{T_{opt}} \cdot u \cdot \frac{d_{50}}{d}$.
Doing so, the combined equation becomes $R_{bio}^2 = const. \cdot k$ or $R_{bio} = const. \cdot \sqrt{k}$. Equation 12 is the result of linear regression between $R_{bio}$ and $\sqrt{k}$.

*Why would burrowing organisms respond like this?*
> Possible explanations for the behavior of species acting at the sediment surface are given in section 5.2.

*How can eqns 11,12 be correct (or an un-biased phenomenological relationship) if all but one of the data points are above the functional curve (Fig 12)?*
> Eq. 12 was presented without a constant offset, therefore it only contains the gradient (or slope) in the data. A constant value would provide a better match with the observations. We will change the fit between biogenic overturning and physical parameters according to the following comments.

*Another argument why it can't be correct is that the 'constant' 3.524 has units (mˆ0.5 sˆ-0.5 - ignoring the mess of m vs mm, sec vs day for simplicity), and hence contains part of the processes.*
> Eq. 12 was constructed to include **all** individual relations (and processes). The units are the result of the root function.

*A proper phenomenological function with should have fitted constants that are non-dimensional.*
> We agree that a dimensionless approach is better to express the relations. A part of the individual physical quantities evaluated ($T$, $u$, $d_{50}$) could be 'summarized' in the non-dimensional particle Reynolds number:
$Re_p = \frac{\rho \cdot u \cdot d_{50}}{\mu}$ (Fig. 1).

*What, in the end, is the physical/biological meaning of eqn 11,12?*
> Possible explanations for the relation between biogenic and the physical boundary conditions were given in section 5.2. The relations in eq. 11/12 were meant to 'summarize' these effects.

*Why use current velocity, and not current shear stress and wave shear stress which were both shown to be important earlier on?*
> We chose current velocity as the most simple parameter describing this aspect of the physical forcing. It was the measured parameter. All later calculations of shear stresses would just mimic this.

*This should all be re-done, using a uniform set of units, starting from relationships that make biological/physical sense, and using dimension analysis to plug the gaps, and using multi-variate regression if/where appropriate.*
> We agree that the consistent use of SI units is better for this approach. The regressions will be repeated using multivariate analysis as the parameters are not indepen-

dent (e.g. lower flow velocities and smaller median grain sizes at the deeper stations). We will provide dimensionless equations in a reviewed manuscript.

*I am not sure if this kind of approach is realistic and feasible, also given the relatively few data points that the authors have. One option they could consider is to abandon this approach and do something more feasible with the data?*
> This is a presentation of a unique dataset. We provide this data and an interpretation of it, and show how the data relates to physical boundary conditions. We think that this is a common and correct way of research. Providing data will allow others to find better, 'more feasible' answers.

———————————————————

[Figure]

$$Re_p = \frac{\rho \cdot u \cdot d_{50}}{\mu}$$

**Fig. 1.** Correlation of the overturning rates with the particle Reynolds number.